# Prognostic Significance of Blood-Based Baseline Biomarkers in Treatment-Resistant Depression: A Literature Review of Available Studies on Treatment Response

**DOI:** 10.3390/brainsci12070940

**Published:** 2022-07-18

**Authors:** Theano Gkesoglou, Stavroula I. Bargiota, Eleni Iordanidou, Miltiadis Vasiliadis, Vasilios-Panteleimon Bozikas, Agorastos Agorastos

**Affiliations:** 1II Department of Psychiatry, School of Medicine, Aristotle University of Thessaloniki, Lagkada Str. 196, 56430 Thessaloniki, Greece; gtheano@yahoo.com (T.G.); stavroula.bargiota@gmail.com (S.I.B.); eleni.iord@gmail.com (E.I.); miltosvasiliadis@gmail.com (M.V.); mpozikas@auth.gr (V.-P.B.); 2VA Center of Excellence for Stress and Mental Health (CESAMH), VA San Diego Healthcare System, La Jolla, San Diego, CA 92161, USA

**Keywords:** major depressive disorder, treatment-resistant depression, antidepressants, biomarkers, treatment response

## Abstract

Major depressive disorder is a leading cause of disability worldwide and a major contributor to the overall global burden of disease. While there are several options for antidepressant treatment, only about 40–60% of patients respond to initial monotherapy, while 30–40% of patients may even show resistance to treatment. This article offers a narrative review of those studies evaluating the predictive properties of various blood-based baseline biomarkers regarding treatment responses to the pharmacological, stimulation, or behavioral treatment of patients with treatment-resistant depression (TRD). Our results show that overall, there is only a very limited number of studies assessing baseline peripheral biomarkers regarding treatment response in TRD. Although there is some evidence for the predictive significance of particular biomarkers (e.g., IL-6, CRP, BDNF), the majority of the results are either single-study reports or studies with conflicting results. This may contribute to the wide variety of treatment protocols and different TRD definition criteria, the small number of patients included, and the existence of different biological phenotypes of the disorder used within the various studies. Taken together, there does not yet appear to be any specific baseline peripheral biomarker with sufficient discriminative predictive validity that can be used in the routine clinical practice of TRD. The discovery of new biomarkers and the better clinical characterization of known biomarkers could support the better classification and staging of TRD, the development of personalized treatment algorithms with higher rates of remission and fewer side effects, and the development of new precision drugs for specific subgroups of patients.

## 1. Introduction

Major depressive disorder (MDD) is a serious disorder with high prevalence, affecting at least 350 million people worldwide and represents the world’s leading cause of disability due to the chronic functional impairment [1,2,3]. MDD includes a heterogeneous spectrum of clinical entities with common features, including emotional symptoms, depressive ideation, loss of energy and disturbed biological functions. These features are often accompanied by several physical and cognitive deficits that additionally affect individual functionality [4]. In particular, a high clinical severity and/or long duration of MDD episodes can lead to critical presentations of the distinct symptoms of the disorder, such as suicidality [5]. The World Health Organization (WHO) estimates that more than 700,000 people commit suicide every year [6]. Additionally, as a recurrent and chronic disorder, MDD is commonly associated with a significantly higher risk for chronic physical comorbidities, resulting in higher total mortality rates [1,2]. All the aforementioned facts underline the urgent need for further actions promoting early prevention, proper diagnosis, and more efficient and individualized treatment strategies for MDD [7,8].

Along with psychotherapy [9], pharmacotherapy with antidepressant agents is considered the best-established, most effective and widely-used first-line intervention for MDD treatment to date [2,10]. Despite the numerous different classes and individual agents of antidepressant medication with proven efficacy, most studies have repeatedly shown that only about 40–60% of patients respond to an initial antidepressant monotherapy treatment, while only one-third reach complete remission [11,12,13]. More importantly, a large proportion of all MDD patients may be resistant to treatment, even under an optimal antidepressant treatment regime according to international guidelines [13].

### 1.1. Treatment Resistant Depression

Treatment-resistant depression (TDR) accounts for approximately 30–40% of patients with MDD and is related to a large direct and indirect societal financial burden that represents up to 70% of MDD’s total cost [14]. Partial- or non-responsiveness to antidepressive treatment contributes to disease chronicity, poor quality of life and lower productivity, leading to a significant increase in healthcare expenses, as well as higher relapse rates and suicide risk [2,15]. Patients with TRD visit general practitioners seven times more often and have three times longer durations of hospitalizations than MDD patients [16]. Impressively, the annual cost of TRD in the U.S.A. alone is estimated at 44 billion dollars [17].

The term TRD was first introduced 50 years ago [18]. However, there has been no unanimous consensus reached with respect to the exact definition of TRD [13,19,20,21,22]. For instance, a recent systematic review mentioned over 150 different definitions of TRD in the relevant literature [22]. Currently, several ongoing studies focus on the further clarification and formulation of an accurate, clinically relevant, and scientifically-sound definition of TRD [20,21,22]. Nevertheless, the most widely accepted clinical research definition to date classifies TRD as a poor response to two or more antidepressant treatments of adequate dose and duration during the current depressive episode [19,20,21,22]. In accordance with this, most real-world studies consider MDD patients resistant to treatment after having an insufficient response to at least two types of antidepressants [23]. Other studies, though, have used different criteria such as resistance to at least three antidepressant agents, a poor response to ketamine, or other definitions based on scoring cut-offs using various psychometric tools. A new publication using a Delphi-method-based consensus approach to define TRD recently provided recommendations for definition and operational criteria for future clinical studies [24].

In addition to its debatable definition and clinical diversity, the inefficient treatment of TRD, which still lacks robust evidence-based grounds, continues to reflect important flaws in the scientific understanding of the disorder [25]. Even the international guidelines are ambiguous about the most appropriate therapeutic approach for each patient at any given step of the therapeutic algorithm [26]. Thus, the treatment of TRD patients is usually based on the personal clinical experience of the treating physician [27]. Different strategies on dosage titration and the switching of antidepressants and their combination with additional antidepressants or other pharmacotherapeutic agents (e.g., mood stabilizers or antipsychotics) are often used, while every further step of the therapeutic algorithm shows reduced efficacy [12,26,28]. Despite the partial improvement of symptoms, psychosocial and working functionality may, however, remain significantly affected, while substantial side effects might emerge, increasing the rates of poor treatment compliance [26,29].

Therefore, the development of clinically-reliable prognostic biological parameters for treatment response is crucial in order to establish personalized and targeted therapeutic algorithms with rapid action and higher efficacy toward TRD. Decades of clinical and experimental research in this direction have laid the foundations for achieving this goal [30,31,32]. However, apart from certain repeatedly confirmed clinical and psychosocial predictive factors of non-response [33], there is a clear lack of objective, personalized, and clinically applicable biomarkers that are potentially able to detect depressed patients with TRD and different responses to antidepressant treatment and/or predict the optimal treatment strategy in every day clinical practice [30,34]. This issue is also reflected in the fact that no biomarker has been included in any way in the definition of TRD to date [19].

### 1.2. Biomarkers in Psychiatry

A biomarker is defined as a biological parameter that can be measured and objectively evaluated as a marker of normal biologic processes, pathophysiologic deviations, or responses to treatment [35]. Despite the increasing research interest in biomarkers for psychiatry, very few have been established in clinical practice, and many findings remain unconfirmed [36,37]. Most of them refer to functional biomarkers (i.e., biomarkers that are not stable and can change during treatment), while especially prognostic biomarkers of treatment response have been of particular clinical significance in the field of psychiatry [38] and necessary for the introduction of any personalized treatment approach.

When comparing all types of possible biomarkers (e.g., imaging, histologic, physiologic, peripheral, etc.), the peripheral ones (e.g., blood, saliva, urine, cerebrospinal fluid, and hair) represent the most accessible biomarkers, taking into account their aptness and ease of collection in every-day clinical practice [39]. In addition, peripheral samples usually provide high material concentrations and quality and also offer the opportunity for simultaneous assessments of several different biomarkers in parallel (e.g., molecular, endocrine, biochemical/metabolic, cytologic, immunologic, genetic, and epigenetic), which is very important in the search for functional and prognostic biomarkers [31,32].

In patients with MDD, numerous studies have shown molecular, endocrine, and functional changes in several central homeostatic systems, including the central nervous system (CNS), the hypothalamic-pituitary-adrenal (HPA) axis, the autonomic nervous system (ANS), and the metabolic and the immune systems [31]. Accordingly, most clinical studies that assess treatment response biomarkers also focus on these systems [40]. However, with respect to TRD patients, similar studies on the predictive biomarkers of treatment response are relatively sparse and display a large number of practical, clinical, and research difficulties.

## 2. Objectives & Methods

In the current review, we systematically present the state of knowledge on prognostic, blood-based, baseline (i.e., before adjunctive treatment initiation) peripheral biomarkers (excluding epigenetic and genetic biomarkers) for responses to adjunctive treatment (e.g., pharmacological treatment, electroconvulsive treatment, and neurostimulation) in TRD. Medline searches were performed using the keywords “treatment-resistant depression”, “biomarker”, “treatment response”, and “prediction”, as well as different combinations of these as search terms. The available literature from 1970 onwards until January 2022 was screened for relevance, and additional material was added from the bibliography of the qualified papers. Papers presenting ‘only the associations of biomarker levels with symptoms’ or ‘only biomarker level changes in course of therapy’ but with no predictive value of baseline biomarker levels toward treatment response were not included in this review. The remaining studies could not be approached via a systematic review process because of the highly diverse definitions of TRD and the variations in methodology, adjective treatments, and the timeframes used in the different studies. Therefore, the literature is presented as a narrative review, providing an overview and discussion of the most important and clinically relevant findings and their prognostic value in every-day clinical practice (cf. Table 1).

## 3. Peripheral Prognostic Biomarkers in TRD Treatment

### 3.1. Immune and Inflammatory Biomarkers

The “immune hypothesis” of MDD suggests a strong link between MDD and a dysfunctional immune system, while mounting research data indicate the significant role of several pro-inflammatory pathways in the pathophysiology of the disorder [68,69,70]. Accordingly, both immune and inflammatory biomarkers have been extensively studied with respect to their potential for predicting responses to antidepressant treatment. For example, several meta-analyses have repeatedly indicated that levels of tumor necrosis factor-alpha (TNF-α), interleukin-6 (IL-6), IL-1β, brain-derived neurotrophic factor (BDNF), IL-8 and c-reactive-protein (CRP) in peripheral blood serum are reliable biomarkers of antidepressant treatment response in MDD [71,72]. More specifically, higher baseline levels of TNF-α, IL-6, BDNF, IL-1β, and CRP, as well as lower IL-8 levels in the blood, are related to poorer responses to pharmacotherapy in MDD. However, there are also studies that do not confirm these results [31,37,38,40,71,72,73], while especially in TRD, respective research addressing baseline immune and inflammatory biomarkers for the prediction of treatment response is relatively scarce.

#### 3.1.1. IL-1β

We have identified three studies regarding the assessment of treatment response prediction in TRD with respect to IL-1β baseline levels. In an open study of ketamine infusion as monotherapy, in a sample of 16 TRD patients, Yang et al. reported significantly higher IL-1β serum baseline levels, as well as significantly reduced IL-1β levels 230 min and 1 day after ketamine infusion in responders compared to non-responders [41]. In contrast, in a similar study of 33 TRD patients, Kiraly et al. did not find any correlation between IL-1β baseline levels and treatment response to i.v. ketamine as monotherapy [47]. Similarly, an open study of add-on electroconvulsive therapy (ECT) in a sample of 29 patients with TRD by Kruse et al., also showed no correlation between IL-1β baseline blood levels and response to treatment [42], suggesting that IL-1β baseline levels have no prognostic value for treatment response in TRD.

#### 3.1.2. IL-6

With respect to IL-6, there is a better line of evidence concerning its predictive value, as several studies have reported an association between plasma IL-6 and treatment response in TRD, while a recent systematic review by Yang et al. concluded that higher baseline levels of IL-6 predicted better responses in patients with TRD, although several contradictory results were reported [74]. In particular, a study by Chen et al. assessed the add-on use of intravenous ketamine in TRD patients and pointed out a connection between higher baseline levels of IL-6 and a better response to therapy in the group subjected to 0.5 mg/kg ketamine infusion but different results in the groups receiving 0.2 mg/kg ketamine and a placebo [43]. Similarly, Yang et al. also noted significantly higher baseline levels of IL-6 in TRD responders to 0.5 mg/kg ketamine infusion monotherapy compared to the group of non-responders [41], while in the ECT add-on study by Kruse et al., higher levels of IL-6 prior to treatment also predicted lower scores in the Montgomery–Asberg Depression Rating Scale following a course of treatment in females, but not in men [42]. In addition, in an add-on study of metyrapone on SSRI treatment in 63 TRD patients, Strawbridge et al. indicated a correlation between higher baseline levels of IL-6 and poorer responses to treatment [49].

On the other hand, Kiraly et al. didn’t discover any prognostic association between IL-6 levels and responses to treatment with 0.5 mg/kg ketamine infusion monotherapy in TRD patients [47]. Similarly, Kagawa et al. found no correlation between baseline IL-6 levels and clinical responses in augmentation therapy with lamotrigine in TRD patients, nor between baseline IL-6 levels and improvements in MADRS score [44]. Allen et al. also failed to find any association between baseline IL-6 levels and responses to treatment with 0.5 mg/kg ketamine infusion or ECT monotherapy in TRD patients, although a correlation between the greater decrease of depressive symptoms and higher baseline levels of IL-6 was found only in the first 24 hours post ketamine infusion [45]. Finally, Yoshimura et al. concluded that baseline levels of IL-6 in the blood had no prognostic value with respect to treatment response in patients resistant to therapy with SSRIs/SNRIs [48].

#### 3.1.3. IL-8

With respect to the predictive significance of baseline IL-8 levels for treatment responses in TRD, we could identify only three studies; of these, one study concluded that higher baseline levels of IL-8 were related to less serious depressive symptoms after the third infusion of 0.5 mg/kg ketamine in patients with TRD but not for other time points during treatment [45]. The two other studies could not report any prognostic correlation between baseline IL-8 levels and responses to treatment under 0.5 mg/kg ketamine infusion monotherapy [47] or ECT add-on treatment [42].

#### 3.1.4. IL-10

Only three studies were found that investigated the prognostic value of baseline IL-10 levels on treatment responses in TRD patients. Allen et al. [45] and Kiraly et al. [47] both could not show any correlation between baseline IL-10 and responses to treatment with 0.5 mg/kg ketamine infusion monotherapy, while Strawbridge et al. also reported no prognostic value of baseline IL-10 on treatment responses in an add-on study of metyrapone on SSRI treatment in 63 TRD patients [49].

#### 3.1.5. IFN-γ

The two studies by Allen et al. [45] and Kiraly et al. [47] were the only studies found to investigate an association between baseline INF-γ levels and treatment response to 0.5 mg/kg ketamine infusion monotherapy in TRD patients, and both failed to find any significant prognostic correlations, which is also supported by missing supportive data for INF-γ in the systematic review of Yang et al. [74].

#### 3.1.6. TNF-α

Several studies have assessed the prognostic value of baseline TNF-a levels in blood and treatment responses in TRD; however, most of them failed to report any prognostic correlations. For example, Chen et al. found that baseline TNF-a levels were not associated with treatment outcomes for both 0.5 and 0.2 mg/kg i.v. doses of add-on ketamine treatment in 47 TRD patients [43]. Similarly, the two studies by Yang et al. [41] and Kiraly et al. [47] on the treatment response of TRD patients to 0.5 mg/kg ketamine infusion monotherapy did not find differences in baseline TNF-a levels between responders and non-responders to 0.5 mg/kg ketamine infusion monotherapy, and could not report any prognostic correlation between baseline TNF-a and treatment response. In addition, no prognostic correlation between baseline TNF-a levels and treatment response was found in studies assessing the response to ECT add-on treatment [42], SSRI/SNRI add-on therapy [48], and metyrapone add-on treatment to SSRIs [49] in patients with TRD. The systematic review oby Yang et al. also did not find any supporting data for the prognostic value of TNF-a in clinical response trials in TRD [74]. However, the experimental trial by Raison et al., assessing the treatment response of TRD patients with mild resistance to treatment with the functional TNF-a antagonist infliximab, was the only study that could show that higher baseline TNF-a levels were associated with a better treatment response [50].

#### 3.1.7. CRP

A large number of studies have assessed the predictive value of baseline peripheral CRP levels within TRD treatment response, and the recent systematic review by Yang et al. actually supports a clinically significant prognostic association between higher CRP reference levels and a better treatment response [74]. For example, the study by Raison et al. found that baseline levels of hsCRP > 5mg/L predicted a larger decrease in Hamilton (HAMD)-17 scale scores in 30 TRD patients with mild resistance to treatment receiving infliximab treatment [50], while Papakostas et al. also reported a larger decrease of HAMD-17 scores in SSRI-resistant depressive patients with higher levels of hsCRP receiving adjunctive therapy with L-methylfolate [51]. However, some studies have shown conflicting or negative results. For example, no prognostic correlations between baseline peripheral CRP levels and treatment response were found in studies assessing the response of TRD patients to both 0.5 and 0.2 mg/kg i.v. doses of add-on ketamine [43] or metyrapone treatment [49]. Interestingly, in their ECT add-on study, Kruse et al. pointed out that baseline CRP levels correlated significantly with final MADRS scores at the end of treatment in the female TRD sample only, while no correlation could be found with respect to the changes in the MADRS score in the total sample of patients [42].

#### 3.1.8. BDNF and Other Growth Factors

Growth factors, like brain-derived neurotrophic factor (BDNF) have been often investigated as response biomarkers of depression treatment and especially implicated in the rapid mechanism of action of ketamine [75,76]. In addition, BDNF is considered to play an important role in the neuroimmune and inflammatory pathophysiology of MDD [77,78]. However, to date, only a few studies have managed to show some correlation between BDNF and treatment response in TRD.

For example, the ECT add-on study of Piccini et al., in 18 patients with TRD, reported lower baseline BDNF levels in patients vs. the control subjects, an increase in BDNF after treatment and, most importantly, higher BDNF baseline levels in responders than in non-responders [52]. Similarly, Haile et al. investigated 22 patients with TRD and found that i.v. ketamine monotherapy resulted in a higher BDNF increase in the serum of the responders than in the non-responders, as well as a negative correlation between MADRS scores and BDNF levels [54]. However, there was no significant prognostic correlation between BDNF baseline levels and responses to treatment with ketamine [54]. On the other hand, a study of treatment responses to add-on riluzole or a placebo in 55 TRD patients by Wilkinson et al. reported lower baseline levels of BDNF in responders to riluzole or the placebo compared to non-responders, although the statistical significance remained within the trend level [53].

Nevertheless, most studies assessing baseline peripheral BDNF could not show any correlation with respect to treatment response. For instance, in an add-on ECT response study of 74 patients with TRD, Maffioletti et al. could not show any difference between the baseline BDNF levels in responders and in non-responders [55]. Similarly, Huang et al. also showed no correlation between the baseline levels of BDNF in serum and responses to treatment in a comparison study of ECT vs. anesthesia with ketamine and propophol in 30 TRD patients [56], as did Allen et al. in a monotherapy study with either ECT or 0.5 mg/kg i.v. ketamine infusion in a group of 35 patients with TRD [46]. Likewise, Kagawa et al. reported only minor variations of BDNF baseline levels between responders and non-responders in a study of adjunctive treatment with lamotrigine in 46 TRD patients, while there was no prognostic correlation found between baseline levels of BDNF and responses to treatment, nor any important association between BDNF-level changes and the improvement of MADRS scores [44].

There have been few studies that managed to indicate a correlation between responses to treatment of TRD and other baseline growth factors. For example, Pisoni et al. found that among all growth factors studied, only vascular endothelial growth factor-C (VEGF-C) showed any prognostic correlation between the treatment response of a group of 36 patients and TRD. In this case, lower baseline levels before treatment were related to better responses to an add-on antidepressant treatment [57]. In another response study of add-on repeated transcranial magnetic stimulation (rTMS) in 15 patients with TRD by Fuduka et al., responders to the treatment showed higher initial concentrations of VEGF, while the percentage change in VEGF levels after treatment showed a statistically-significant correlation with the changes in psychometric scores of depressive symptomology [58]. Finally, in an i.v. 0.5 mg/kg ketamine infusion monotherapy study of 33 patients with TRD, Kiraly et al. showed that, among a large number of assessed biomarkers, only baseline serum levels of fibroblast growth factor 2 (FGF-2) were associated with treatment response, where low initial levels predicted better response to treatment [47].

### 3.2. Endocrine Biomarkers

MDD is considered a stress-related disorder with a unique pathophysiological neuroendocrine profile, presenting distinct changes in activity and reactivity of the HPA axis [2,79]. The most consistent findings include correlations between: the hyperactivity of the HPA axis and increased cortisol (CORT) levels, a higher corticotropin-releasing hormone (CRH) drive and higher adrenocorticotropic hormone (ACTH) and vasopressine (AVP) levels, and a weak cortisol awakening response (CAR) and the reduced sensitivity of glycocorticoid receptors [80]. It has also been shown that effective antidepressive treatment with reductions in depressive psychopathology is often related to a consequent normalization in the HPA-axis (re-)activity [80]. Apart from the baseline and diurnal levels of several HPA axis hormones (CRH, AVP, ACTH, CORT, and dehydroepiandrosterone—DHEA), neuroendocrine suppression or stimulation tests (e.g., dexamethasone, metyrapone) have often been used to study dynamic endocrine levels with respect to their possible prognostic significance in antidepressant response [80]. Nevertheless, the studies that investigated the baseline and diurnal activity, and dynamic responsiveness of the HPA axis in TRD, are proportionally scarce.

In one of the three available studies, Markopoulou et al. measured baseline CORT and DHEA and their ratio in 28 patients with TRD, noting their association with treatment responses to add-on pharmacological treatment [59]. In this study, while CORT levels were lower after the treatment, there was no correlation between baseline and post-treatment CORT levels and treatment outcome. On the contrary, the responders to treatment had significantly lower DHEA and showed a higher CORT/DHEA ratio both at admission and discharge compared to the non-responders, suggesting that, although remaining stable across treatment, the CORT/DHEA ratio could represent a prognostic biomarker of response to TRD. In another study, Dinan et al. examined therapeutic responses to add-on therapy using dexamethasone in 10 patients with SSRIs resistance and showed that higher baseline CORT levels predicted treatment response [60]. On the other hand, in a study of treatment responses to add-on sleep deprivation or sleep phase shift in 21 patients with TRD (unipolar and bipolar), Kurczewska et al. noticed that baseline CORT levels were significantly lower in responders than in non-responders [61].

### 3.3. Metabolic Biomarkers

#### 3.3.1. Adipokines

Adipokines are cytokines, with hormonal action secreted by the lipid tissue (e.g., adiponectin, leptin, and resistin), are considered to participate in the pathophysiological pathways connecting obesity with cardiovascular diseases. In a 0.5 mg/kg i.v. ketamine infusion add-on study in 8 patients with unipolar/bipolar TRD, Machado-Vieria et al. showed that apart from the prognostic value of BMI, regarding the response to treatment with ketamine, lower baseline concentrations of adiponectin in the serum could predict antidepressant responses to ketamine [62].

#### 3.3.2. Lipidemic Factors

Blood lipidemic factors have been implicated in MDD pathophysiology and especially in intra- and inter-neuronal functioning, and have been studied as biomarkers in MDD [81]. In TRD, only two studies have been found that assess blood lipidemic factors in relation to treatment response. In an add-on study with infliximab in 26 patients with TRD, Bekhbat et al. found that baseline levels of cholesterol, LDL, and non-HDL were higher in responders and also showed significant decreases (during treatment) in those patients with higher baseline CRP [63]. On the contrary, in a study of 92 TRD patients, Papakostas et al. reported that baseline levels of cholesterol > 200 mg/dL predicted a worse response to monotherapy with nortriptyline [64].

### 3.4. Other Biomarkers

#### 3.4.1. Protein p11

The S100 calcium-binding protein A10 (S100A10), also known as protein p11, belongs to a family of proteins that regulate a number of cellular processes, such as cell-cycle progression, differentiation, exo-/endocytosis, and the transport of neurotransmitters. Protein p11 has been noted to be involved in the pathophysiology of mood disorders, as well as in the ketamine mechanism of action, and has been suggested to be a biomarker of response to therapy with also other antidepressants [82]. The only study found that addressed the predictive role of p11 in TRD response was conducted by Veldman et al. [65]. This study investigated baseline p11 levels in association with the responses of 30 patients resistant to SSRIs given 0.5 mg/kg i.v. ketamine infusion monotherapy, and showed that higher baseline levels of p11 in cytotoxic T-lymph cells were correlated with better responses to treatment.

#### 3.4.2. D-Serine

D-serine is an endogenous co-agonist at the “glycine site” of NMDA receptors and has been implicated in the pathophysiology of depression, as it plays a significant role in NMDA neurotransmission and neuroplasticity [83]. In the only identified study assessing the prognostic significance of baseline D-serine levels with responses to monotherapy with 0.5 mg/kg i.v. ketamine infusion (in 21 TRD patients), Moaddel et al. reported that baseline D-serine plasma levels were significantly lower in the responders than in the non-responders, while lower baseline D-serine plasma levels predicted the antidepressant responses to ketamine and was able to explain 60% of the variance in the clinical responses [66].

#### 3.4.3. Oxidative Stress Biomarkers

Increased oxidative stress is believed to play a catalytic role in the multisystemic etiopathology of MDD [84]. We have identified only one study assessing baseline oxidative stress status with regard to treatment responses in TRD. In this study, Stirton et al. studied the response of 48 TRD patients to add-on rTMS and reported a significant correlation between responses to treatment and higher baseline levels of oxidized phosphatidylcholine, although they could not show any association with this to baseline oxylipin levels [67].

## 4. Discussion

Major depressive disorder is a highly prevalent disorder and represents the world’s leading cause of disability. However, despite the availability of numerous pharmacotherapeutic alternatives, a large proportion of patients do not properly respond to treatment and may be resistant to treatment. The identification of prognostic biomarkers for treatment response is therefore crucial for establishing a more personalized, targeted, rapid, and efficacious treatment algorithm for MDD. However, there are no easily accessible, reliable, and clinically applicable biomarkers as yet identified, supported by sufficient data, and that could be routinely used in everyday clinical practice. This gap appears to be even bigger concerning the potential biomarkers for treatment responses in TRD.

The present article offers a narrative review of those studies evaluating the predictive quality of various baseline blood-based peripheral biomarkers in treatment response to the pharmacological, stimulation, or behavioral treatment of patients with treatment-resistant depression. Overall, we have found a very limited number of studies assessing baseline peripheral biomarkers with regard to treatment response. Although there is some evidence for the predictive significance of particular biomarkers regarding responses to the treatment of TRD (e.g., IL-6, CRP, BDNF), the majority of the results are either single-study reports or studies with conflicting results. This situation may have additionally complications due to several factors, such as the very small number of included patients in the studies, the very different treatment protocols and study designs used, as well as the different definition criteria of TRD used in the studies.

Another important factor that complicates our overall understanding, as well as the clinical management of TRD, is the distinct heterogeneity of the clinical phenotypes in MDD, suggesting heterogeneous biological backgrounds which lack established diagnostic tests capable of determining pathophysiological subgroups. The use of specific prognostic biomarkers regarding the response to treatment in TRD should accordingly mirror the specific pathophysiological pathway of the specific phenotype in which the biomarkers are involved or the mechanism of the specific applied treatment [85]. For example, the reviewed studies examining anti-inflammatory treatment factors (e.g., infliximab and L-methylfolic acid) found that higher levels of inflammation often predicted the responses to treatment in TRD. On the other hand, most studies using treatments without anti-inflammatory properties (e.g., lamotrigine and SSRIs/SNRIs) found no correlation between inflammation levels and the therapeutic results in TRD, with the exception of one study showing a correlation between baseline IL-6 and worse treatment outcomes with SSRIs/SNRIs [48]. These results point out that inflammatory biomarkers could be clinically useful for treatment prediction in TRD when addressing responses to anti-inflammatory agents or investigating patients with an inflammatory sub-phenotype of depression, with this possibly supporting new intervention strategies [71,74].

Also, taking into consideration the small number of participants in the majority of the reviewed studies, additional and larger prospective trials will be needed to explore the prognosis of treatment responses in TRD. Additionally, the investigation of specific blood-based, peripheral biomarkers should concurrently target factors from different pathophysiological pathways and be combined with (epi-)genetic, psychometric, and environmental factors, acknowledging also the individual medical histories of patients (e.g., immune/metabolic comorbidities and exposure to childhood trauma) in order to help establish sensitive and specific biomarkers for the different biological sub-phenotypes of the disorder [19].

The detection of new biomarkers and the improvement of the clinical characterization of already existing biomarkers in MDD and TRD could be helpful with the following: (i) better biological characterizations of the resistance to treatment, (ii) better staging of the disorder, (iii) better molecular distinctions of the pathophysiological sub-phenotypes, (iv) the development of personalized pharmaceutical algorithms in specific patient subgroups, (v) the improvement of response time and effectiveness of treatments, (vi) the detection of new pharmacodynamic targets in the treatment of TRD and (vii) the minimization of harmful side effects [32,73].

## 5. Conclusions

The non-satisfactory treatment outcomes for TRD uncover the need for an urgent improvement in our therapeutic approach; the establishment of clinically useful, easily applicable, and scientifically accurate prognostic biomarkers for treatment response is needed [37,38]. Our results show that overall, there are only a very limited number of studies assessing baseline peripheral biomarkers with regard to treatment response in TRD and that there does not yet appear to be any specific baseline peripheral biomarker that has sufficient discriminative predictive validity that can be used in the routine clinical practice of TRD. However, some results suggest that the discovery of new biomarkers and the better clinical characterization of known biomarkers could potentially help associate these with the specific biological phenotypes of the disorder, supporting a better classification and staging of depression, the development of personalized-treatment algorithms with higher rates of remission and fewer side effects, and the development of new precision drugs for specific subgroups of patients.

## Figures and Tables

**Table 1 brainsci-12-00940-t001:** Overview of studies assessing the prognostic significance of blood-based baseline biomarkers of responses to treatment in TRD.

Study	Ν	Sample	Treatment	Design	Factors Related with Treatment Response	Relation to Response +/−	Factors not Related with Treatment Response
Yang et al. [41]	16	Unipolar	Ketamine 0.5 mg/kg i.v.	Monotherapy	IL-6, IL-1β	+	TNF-a
Kruse et al. [42]	29	Unipolar	ECT	Add-on	IL-6 (females) CRP (females)	+	IL-6 (males) IL-1β, IL-8, TNF-a, CRP (males)
Chen et al. [43]	47	Unipolar	Ketamine 0.5/0.2 mg/dL i.v.	Add-on	ΙL-6 (Dose: 0.5mg/dL)	+	IL-6 (Dose: 0.2mg/dl), TNF-a, CRP
Kagawa et al. [44]	46	Unipolar/Bipolar	Lamotrigine	Add-on	N/A	N/A	IL-6, BDNF
Allen et al. [45,46]	17/18	Unipolar	Ketamine 0.5 mg/kg i.v./ECT	Monotherapy	N/A	N/A	IL-6, IL-8, IL-10, IFN-γ, CAR, Kynurenine, BDNF
Kiraly et al. [47]	33	Unipolar	Ketamine 0.5 mg/kg i.v.	Monotherapy	FGF-2	-	IL-6, IL-1α, IL-1β, TNF-α, EGF, FLT3L, Fractalkine, G-CSF, GM-CSF, GRO, IFN-2a, IFNr, IL-10, IL-12P40, IL-12P70, IL-13, IL-15, IL-17a, Il-1ra, IL-2, IL-3, IL-4, IL-5, IL-7, IL-8, IL-9, IP-10, MCP-1, MCP-3, MDC, Mip-1a, Mip-1b, PDGF-AA, PDGF-BB, RANTES, scd40L, TGF-α, TNF-β, VEGF
Yoshimura et al. [48]	20	Unipolar	SSRIs/SNRIs	Monotherapy/Add-on	N/A	N/A	IL-6, TNF-a
Strawbridge et al. [49]	63	Unipolar	SSRIs + methyrapone	Add-on	IL-6	-	TNF-a, IL-10, CRP
Raison et al. [50]	30	Unipolar/Bipolar	Infliximab	Monotherapy/Add-on	TNF-a hsCRP	+	N/A
Papakostas et al. [51]	75	Unipolar	SSRIs + L-methylfolate		hsCRP	(+)	N/A
Piccinni et al. [52]	18	Unipolar/Bipolar	ECT	Add-on	BDNF	+	N/A
Wilkinson et al. [53]	55	Unipolar	Riluzole	Add-on	BDNF	(−)	N/A
Haile et al. [54]	22	Unipolar	Ketamine 0.5 mg/kg i.v.	Monotherapy	BDNF	(+)	N/A
Maffioletti et al. [55]	74	Unipolar	ECT	Add-on	N/A	N/A	BDNF
Huang et al. [56]	30	Unipolar	Ketamine 0.5 mg/kg i.v./Propofol 0.5 mg/kg i.v.	Monotherapy	N/A	N/A	BDNF
Pisoni et al. [57]	36	Unipolar/Bipolar	Antidepressants, ECT, Psychological Therapy	Add-on	VEGF-C	+	N/A
Fukuda et al. [58]	15	Unipolar	r-TMS	Add-on	VEGF	+	N/A
Markopoulou et al. [59]	28	Unipolar	Pharmacological Treatment (antidepressants, mood stabilizers, antipsychotics, benzodiazepines, thyroid hormones, buspirone)	Add-on	CORT/DHEA	+	N/A
Dinan et al. [60]	10	Unipolar	SSRIs + dexamethasone	Add-on	CORT	+	N/A
Kurczewska et al. [61]	21	Unipolar/Bipolar	Sleep deprivation/Sleep phase shift	Add-on	CORT	-	N/A
Machado-Vieira et al. [62]	8	Unipolar/Bipolar	Ketamine 0.5 mg/kg i.v.	Add-on	Adiponectin	-	N/A
Bekhbat et al. [63]	26	Unipolar/Bipolar	Infliximab	Add-on	Cholesterol, LDL, non-HDL	+	N/A
Papakostas et al. [64]	92	Unipolar	Nortriptyline	Monotherapy	Cholesterol > 200mg/dL	-	N/A
Veldman et al. [65]	30	Unipolar	Ketamine 0.5 mg/kg i.v.	Monotherapy	p 11	+	N/A
Moaddel et al. [66]	21	Unipolar	Ketamine 0.5 mg/kg i.v.	Monotherapy	D-serine	-	N/A
Stirton et al. [67]	48	Unipolar	r-TMS	Add-on	Oxidized Phophatidylcholine	+	Oxylipins

Abbreviations: N/A, not applicable; +, positive association to response; -, negative association to response.

## Data Availability

Not applicable.

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
