# Peer review of "Prognostic Significance of Blood-Based Baseline Biomarkers in Treatment-Resistant Depression: A Literature Review of Available Studies on Treatment Response"

_brainsci, 2022, doi:10.3390/brainsci12070940_

Round 1

Reviewer 1 Report

·       This is a narrative review of studies evaluating the predictive properties of various blood-based baseline biomarkers regarding treatment response to pharmacological, stimulation or behavioral treatment in patients with treatment-resistant depression. The results show that there is some evidence, however there are several limitations to the individual studies e.g., differing TRD definition, small sample size, and non-comparability. The authors conclude that there is currently no specific peripheral biomarker with sufficient discriminatory predictive validity that could be used in clinical practice. Overall, I found this review informative and a suitable fit for this journal. I have only two minor comments.

·       There is a recent consensus guideline for definition of treatment-resistant depression for clinical trials. This article could be discussed in the introduction as well.

o   Sforzini, L., Worrell, C., Kose, M., Anderson, I. M., Aouizerate, B., Arolt, V., . . . Pariante, C. M. (2021). A Delphi-method-based consensus guideline for definition of treatment-resistant depression for clinical trials. Mol Psychiatry. doi:10.1038/s41380-021-01381-x

·       There is a typo in the introduction on page 1: “In particular, high clinical severity and/or long duration of MDD episodes can lead to critical presentations of the distinct syptoms of the disorder, such as suicidality. [5]”

Author Response

Dear Reviewer,

thank you for your positive remarks on our manuscript. 

With respect to your suggestions/comments:

Comment 1: There is a recent consensus guideline for definition of treatment-resistant depression for clinical trials. This article could be discussed in the introduction as well.

o   Sforzini, L., Worrell, C., Kose, M., Anderson, I. M., Aouizerate, B., Arolt, V., . . . Pariante, C. M. (2021). A Delphi-method-based consensus guideline for definition of treatment-resistant depression for clinical trials. Mol Psychiatry. doi:10.1038/s41380-021-01381-x

Answer: Thank you for this remark. We have followed your suggestion and now discuss the included reference in our introduction.

Comment 2: There is a typo in the introduction on page 1: “In particular, high clinical severity and/or long duration of MDD episodes can lead to critical presentations of the distinct syptoms of the disorder, such as suicidality. [5]”

Answer: Thank you for pointing out the typo, we have corrected this in the revision.

Reviewer 2 Report

The authors of this study summarized clinical studies assessing blood-based baseline biomarkers to predict treatment response in TRD, which is clinically relevant. There is no concern that should be considered in publishing this article. Congratulations!

Author Response

Dear Reviewer,

thank you so much for your positive remarks with respect to our review!

----

No comments were made
